# Recent Advances in High Performance Polymers—Tribological Aspects

**Abdulaziz Kurdi [1,2,]*** and **Li Chang [1,]***

[1]   Centre for Advanced Materials Technology, School of Aerospace, Mechanical and Mechatronic Engineering, The University of Sydney, Sydney, NSW 2006, Australia

[2]   The King Abdulaziz City for Science and Technology, The National Centre for Building and Construction Technology, P.O. Box 6086, Riyadh 11442, Saudi Arabia

*   Correspondence: akur7938@uni.sydney.edu.au (A.K.); li.chang@sydney.edu.au (L.C.)

**Abstract:** High-performance polymer (HPP)-based engineering materials in tribological applications have been under continuous research over the last few decades. This paper reviewed the recent studies on the sliding wear properties of HPPs and their nanocomposites, which are associated with the intrinsic and extrinsic parameters. In particular, the effects of the intrinsic properties of polymer composites (e.g., mechanical properties of the materials and the types of fillers) and external environmental conditions (e.g., service temperature and lubrication medium) on the formation of transfer layers (TLs) were discussed. The latter would govern the overall friction and wear of polymeric materials in sliding against metallic counterparts. In addition, correlations between the basic mechanical properties of HPPs and their sliding wear behavior were also explored.

**Keywords:** high-performance polymers; tribology; nanoindentation; TLs

## 1. Introduction

Tribology, a multi-disciplinary field across mechanical and materials engineering, has been under ongoing exploration to ensure the maximum usage of potential materials where friction and wear are unavoidable. Among different material classes, high-performance polymers (HPPs) and their composites are being investigated by various researchers [1–13] for such applications, owing to their high strength/density ratio as well as the structural integrity. A number of the applications, such as seals and bearings, are mainly focused on the tribological performance of HPPs and their composites, especially in non-lubricated, sliding conditions [2–6]. For instance, the exceptional wear behavior of HPPs (commonly against metallic counterparts) without the requirement of lubricants is desirable in a number of applications, such as the textile and food industries, to avoid contamination problems. Further, HPPs have the ability to dampen shock and vibration with excellent corrosion resistance, making them excellent candidates for aerospace, chemical, and offshore applications.

Up to now, different kinds of polymers, including both thermoplastics and thermosets, are frequently used for various tribological applications [8]. For example, epoxy is a common example of a thermoset, commonly withstanding its fragile and sensitive nature towards microfracture. On the contrary, high performance epoxies, which are normally prepared with high cross-link density, may have high Young's modulus, greater strength, durable bond ability, and outstanding chemical constancy [2–5]. In particular, the mechanical properties, as well as the wear resistance of thermoset composites, can be significantly improved by filling with continuous fibers at high volume contents (> 50 vol %), e.g., using the conventional autoclave moulding technique. Nevertheless, the production time that is required for thermoset-based composites is rather long, due to the curing process. It has been envisioned that thermoplastics will be increasingly used, owing to the more cost-effective

injection moulding with particulate and/or short fiber fillers. As noted, neat polymers are commonly 'blended' with different types of fillers to achieve desirable tribological/mechanical properties and other functions. It is important to state that to modify the properties of polymers using fillers, the right term should be "blended" instead of "reinforced". This has been frequently overlooked in the literature. For instance, polytetrafluoroethylene (PTFE) is often employed as a solid lubricant to improve the friction performance, but it does not have an effect on improving (reinforcing) the mechanical properties. Towards that, Hunke et al. [14,15] recently indicated the benefits of surface functionalized PTFE powders for better reinforcement capabilities in HPPs. Also, the philosophy of incorporating fillers in the polymer matrix resembles that of metal matrix filled with inorganic oxide/carbide/nitride particles [16]. The type, form, and compatibility of fillers all have major roles in overall material performance.

In the literature, polymer composites with various types of fiber-like fillers have been well documented [17–26]. Diverse matrix and fiber materials endow composites with exceptional properties for various applications where the retention of the mechanical properties of materials is of prime concern. However, compared to injection-mouldable neat polymers or particle-blended polymers, a drawback of such fiber-reinforced composites is that components fabricated from these materials generally require relatively lengthy fabrication times and a complex dispersing process to maintain the high length/diameter ratio of the fibers. Moreover, the alignment and orientation of fibers in fiber-blended composites are vital, but not in particle-blended polymers. Rasheva et al. [27] pointed out the advantages of nanoparticle addition in polymer matrices over a short carbon fiber (SCF)-type filler blend, as the orientation of the fiber-type filler significantly influenced the wear behavior of the composites and directional loading during sliding. It is also worthwhile indicating that fiber-like fillers may not be easily used for novel additive manufacturing, such as the selective laser sinter (SLS) three-dimensional (3D) printer.

Over the last few decades, the tribological behavior of nanoparticle-blended HPPs has received more and more attention in the research community [16–21]. The strong mechanical properties of HPPs and their composites have led to the expectation of good wear resistance, even at high temperatures. Indeed, HPPs are presently being applied in sliding machinery components or as tribo-pairs in various fields where low friction and wear are required [28–31]. However, the involved wear mechanisms are not been fully understood, particularly due to the complex physical and chemical interactions in the wear process. During the wear, material removal occurs due to complex thermo-mechanical reactions that are dependent on the polymeric chain structure, the types of polymers or co-polymers (thermoplastic/thermoset), the polymeric functional groups, the tactility, molecular weight, and curing/setting behavior of polymer chains. [28]. Moreover, because of the relatively low cohesive energy of polymers when compared to their pairing materials, such as metals and ceramics, polymers are capable of transmitting and distributing the load effectively across the matrix and blended particles, which results in significant reduction in wear and friction [32–35]. With the additional nanofillers, more complicated tribo-chemical reactions might take place in tribo-contact areas, which yield characteristic transfer layers (TLs). Incorporation of nanoparticles into the polymer matrix makes these TLs more stable and resilient, which may significantly decrease friction and wear. The structure and properties of such TLs differ from system to system and their in-depth understanding is required to understand the fundamental aspects of the tribology of HPPs and their composites. For example, such TLs may offer a boundary lubricant effect in tribo-contact, which allows such HPP composites to be used under high $pv$ (contact pressure, $p$ times sliding velocity, $v$) conditions in unlubricated sliding. It is noted that both the terms, TFL and TL, have been used interchangeably in the literature. However, there is a distinction between them, as pointed out by Bahadur et al. [18]. The term TFL is not appropriate when referring to material being transferred from a soft polymer interface to a harder metal counterpart. In fact, TFL is most appropriately used for continuous and uniform materials, while TL is attributed to either uniform or irregular materials along the wear patch or track. To develop high wear-resistant HPPs, it is important to understand

the formation mechanism of TLs, as well as their dependence on the properties of polymer matrix, the type of fillers, and service conditions, such as temperature and lubrication.

Another important aspect of friction and wear is that they are not only dependent on material properties, but also on the tribo-system where the materials are being used, and are termed as "system responses" [36]. Thus, the environmental conditions, such as lubricant medium, temperature, humidity, vibration, etc. also play big roles towards the overall performance of the system. For example, in liquid medium or at high temperatures, the abovementioned TLs might be partly dissolved or melted and become unstable, which sometimes severely outweighs the benefits that are required for dry sliding conditions. Moreover, liquids, such as water, may be absorbed by the polymers, which interrupt their molecular structures and thus deteriorate their tribological properties. Such 'system responses' of tribology makes it difficult to study the wear behavior of HPPs in isolation without considering their application environments, and an integrated approach is required to get the complete picture. Although it is necessary, a complete description of the tribological behavior of HPPs considering the material system and environment is not yet available. There are gaps in the knowledge related to the effects of environment on the tribological performance of such HPPs, and a fundamental understanding that is related to the role of TLs is absent. Additionally, the most recent progress in this field is not well interconnected in terms of the mechanical properties of HPPs and their tribological aspects. To tackle these concerns, the wear mechanisms of HPPs and the effects of diverse system variables, such as sliding speed, vibration, load, and environment temperature, on friction and wear development were examined based on the evidence offered in the literature. Understanding these will enable the design and selection of tribo-materials for specific applications.

Thus, the aim of this review is to get an overall picture of the tribological performance of HPPs and their composites, in terms of their respective mechanical properties, wear debris, and TLs formation, as well as the effect of the service environment.

## 2. Recent Developments of the Tribological Aspects of HPPs

Having a high heat distortion temperature (also known as heat deflection temperature under load, HDT) is a key feature of HPPs. In practice, HPPs are expected to retain their structural integrity over their continuous service temperature (CST), which is commonly higher than 150 °C by definition [18]. In contrast, commonly used polymers, such as polyethylene terephthalate (PET), usually get deformed below 100 °C, which substantially limits their applications under dry sliding conditions, since the contact temperature can be significantly higher [16]. High performance thermoplastics, such as polyamides, can be melted into complex shapes with high physical-mechanical and thermal properties. In fact, polyamides are the most widely used structural thermoplastics and they are characterized by high impact resistance and resistance to fluctuating loads, gasoline, and oils [1–8,37–49]. On the other hand, heat-resistant engineering thermoplastics for specific purposes include fluoroplastics (polyfluoroolefins, fluoropolymers), some of which suffer from poor mechanical properties [16,20]. Fluoroplastics cannot be used as structural plastics, since their physical and mechanical properties are considerably inferior to other plastics, such as polyamides. The typical tensile strength of fluoroplastics does not exceed 30–40 MPa, with a modulus of elasticity of about 0.4 GPa. Since tribological properties are products of the entire system, to develop HPPs and their composites as potential tribo-materials, their properties, such as chemical resistance, mechanical properties, cohesive strength, and the retention of strength and structural integrity at service temperature, should be taken into consideration, according to the sliding conditions of the system in which these materials are intended to function.

### 2.1. Processing Characteristics of HPP Materials

The high thermal stability, as characterized by HDT, of HPPs normally makes their processing difficult and special instruments are often required. Most HPPs are made based on their heat stability, which makes them moderately expensive [29,33]. HPPs are thus about three to 20 times more expensive than common polymers/plastics [18,35]. Among the various HPPs

that are reported in the literature, the followings are some that are commonly used in structural applications: Polyamides (PA), Polysulfone (PSU), Polyethersulfone (PES), Polyphthalamide (PPA), Poly(*p*-vinylidene) fluoride (PVDF), Polyetherimide (PEI), Poly(*p*-phenylene) sulphide (PPS), Polyether ether ketone (PEEK), Styrene-butadiene copolymers (SBC), Polyketone (PK), Poly(ether ketones) (PEK), Poly(para-phenylene) (PPP), and Polybenzimidazole (PBI). Like normal polymers, HPPs are also made up of repeating units of macromolecules that give rise to long polymeric chain structures in three dimensions. Figure 1 shows the reiterating units of PEEK, PPP, and PBI that build up their respective polymer structures.

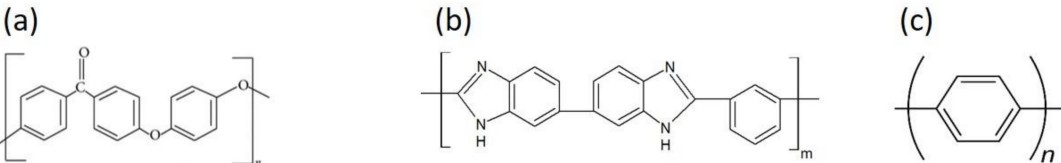

**Figure 1.** Reiterating unit construction of (**a**) Polyether ether ketone (PEEK), (**b**) Poly(para-phenylene) (PPP), and (**c**) Polybenzimidazole (PBI) [50].

Based on the microstructure of polymers, two approaches are usually taken to further enhance their mechanical and thermal performances: (i) the addition of inorganic nanofillers in the form of blends and (ii) co-polymerization by the addition of organic macromolecules. In the latter case, organic macromolecules also serve as "cross-linking" agents. Such macromolecules sometimes incorporate aromatics rings, which, during sliding, offer enhanced resistance against polymer chain movement, even under high contact temperature, and thus retain strong mechanical properties [51]. Other commonly used cross-linking agents include $SO_2$ and CO. By mixing these different compounds, diverse HPPs have been created with different characteristics [16–28]. According to the literature [52–56], a maximum temperature resistance of about 260 °C can be achieved with fluoropolymers, although their wear resistance may not be favorable. Mixture of co-polymers gives rise to amorphous and semi-crystalline polymers: PSU, PES, and PEI are examples of amorphous structures, whereas PPS, PEEK, PBI, and PPP are semi-crystalline. Semi-crystalline polymers can be used even above their glass transition temperature ($T_g$), another added advantage against chemical constancy [54]. Various inorganic nanofillers blends, e.g., silicon nitride ($Si_3N_4$) [45], silicon dioxide ($SiO_2$) [49], goethite ($\alpha$-FeOOH) [19], zirconium dioxide ($ZrO_2$) [21,32,47], and titanium dioxide ($TiO_2$) [32,45,48], have been proved to not only contribute towards enhancing mechanical properties, but also to lowering the friction coefficient and the rate of wear under various sliding circumstances [57–59]. In particular, PEEK, PPS, and PTFE are the most widely studied polymers for different tribological applications and they are often blended with $TiO_2$, SiC, $Si_3N_4$, and carbon fiber fillers. Nevertheless, it is also noted that there are no single or combined polymers or fillers that provide the best tribological performance in all conditions. Being a result of "system responses", friction and wear always depend on both the intrinsic material properties and the external environmental conditions.

*2.2. Tribological Characteristics of HPP Materials and Their Dependence on PV Factor*

In general, the wear of HPPs under sliding is subjective to certain contact circumstances, depending on the bulk mechanical behavior and surface profile of HPPs, as well as TLs. The effect of lubricants and the atmosphere on the wear of a polymer is described by the chemical interactions when it experiences with its contacts. The wear resistance of HPPs can be effectively improved with blended fillers (either nanoparticles or traditional fiber fillers). To predict the wear life and rank the wear resistance of materials, time-related depth of wear rate ($W_t$) is commonly used, which is defined according to Equation (1) [4]:

$$W_t = k^* pv = \frac{\Delta h}{t}(m/s) \tag{1}$$

where $k*$ is the wear constant, $p$ is pressure, $v$ is sliding speed, $t$ is the duration of the test, and $\Delta h$ is the loss of height of the specimen. The wear constant $k*$ is theoretically a material variable that is equivalent to the adjustment of the product of $p$ and $v$. According to Equation (1), $pv$ might be assumed as a tribological measure of the load-carrying ability of materials, which leads to two evaluation variables [28]: (i) basic wear constant $k^*$, which remains unchanged in a specific limit of $pv$ and (ii) limiting $pv$, beyond which the rise of wear rate is excessively fast to be meaningfully used in practical applications. Diminishing the basic wear factor $k^*$ and boost limiting the $pv$ value are the common goals in designing wear-resistant HPP materials.

In practice, for the convenience of material selection, it is commonly assumed that when the limiting $pv$ value is not exceeded the specific wear rate, $k^*$ and the friction coefficient are material parameters and are independent of the $pv$ factor. Nevertheless, to develop new engineering tribo-materials, it is important to systematically evaluate their wear behavior under different $pv$ conditions and to understand the effect of $pv$, as well as the load-carry capacity of the material under the given environmental conditions [60–65]. Pei et al. [1] observed that, with the rise of $pv$ parameters, a general trend of increasing temperature occurred, and it thus concluded that the temperature constancy of polymers would the focus in situations where wear and friction are vital matters. According to Briscoe et al. [66], during the sliding process, polymeric materials can experience very high temperatures that are mainly concentrated at the interface region (within depth of around 100 nm). However, the rest of the material, except the contact region, are at testing temperature. This high temperature in the contact region is significant for the tribological features of polymers, as structural failure of the components often starts from this region.

Research has shown that the increase of velocity mostly affects the tribological properties of polymers by increasing the contact temperature, whereas higher-pressure conditions may change the wear mechanism in different ways, depending on the thermal-mechanical properties of the polymers. In general, with the increasing pressure, the friction coefficient tends to decrease, which can be explained by the thermal/stress softening effect on polymers. On the other hand, the polymer surface could be plasticized under high pressure conditions, which assists in the separation of the surface materials and contributes to acute wear. Zhang et al. [60] noted that, under higher applied pressure, the wear properties of amorphous PEEK might be closely associated with its viscoelastic behavior, which is absent in low load conditions. However, it is worth indicating that such a correlation between wear and intrinsic material properties remains challenging. Based on studies of PBI, Friedrich et al. [28] concluded that the comparative action of hardness or modulus vs. impact toughness or ductility stays vague and more work is required to achieve a good understanding.

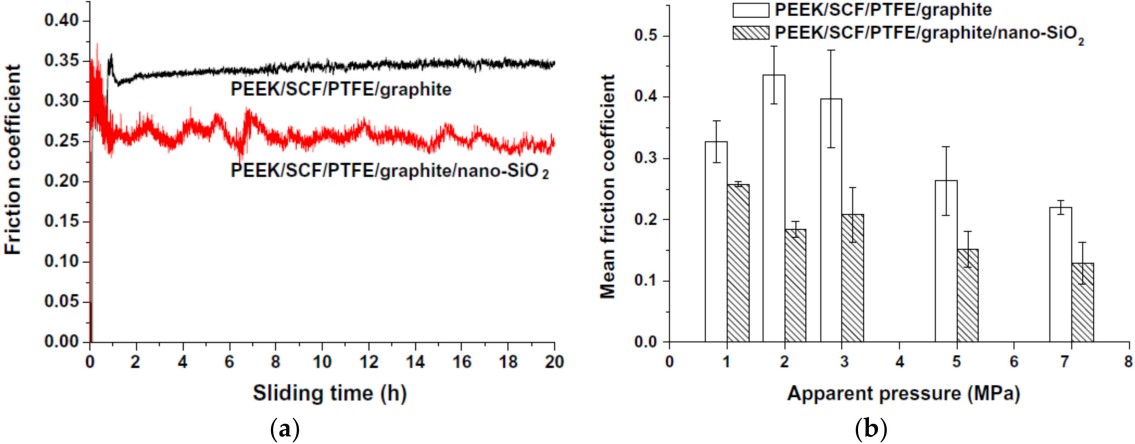

**Figure 2.** (**a**) Typical evolution of friction coefficient as a function of sliding time for PEEK composites filled with and without nanoparticles and (**b**) effects of apparent pressure on the mean friction coefficient in the steady-phase [67].

For HPP composites blended with different fillers, some work showed that the wear rate could be less dependent as a function of the normal load until a critical value was reached [67]. Figure 2 shows the effect of nanofiller (SiO$_2$)-incorporated PEEK composite as a function of pressure and sliding speed [67]. As given in (Figure 2a), under 1 MPa applied pressure, the friction coefficient reaches the stable state rapidly in the case of nanofiller-incorporated composites. After that, there are fluctuations in the evolution of the coefficient of friction, which is absent in case of neat PEEK. This implies the effect of wear debris on the friction coefficient. In all cases (Figure 2b), the mean friction coefficient is almost half of the neat sample.

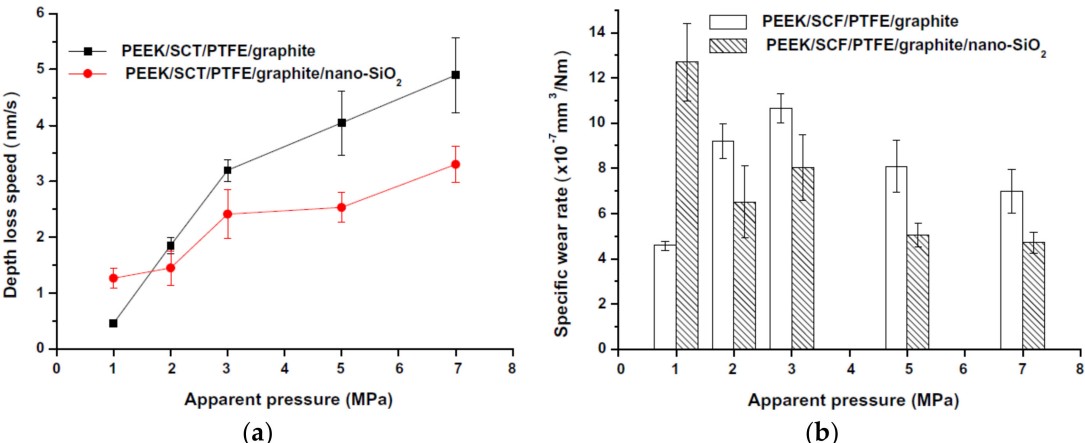

**Figure 3.** Effects of apparent pressure on (**a**) the depth loss speed and (**b**) specific wear rate of PEEK composites filled with and without nanoparticles [67].

The wear behavior of PEEK composites, as shown in Figure 3, shows mixed behavior. Under high pressure and sliding speed, de-bonding of fillers from the matrix takes place, which increases the wear rate. In addition, detached fillers can graze the matrix material and lead to further removal of materials as sliding continues. The grazing effect of broken fillers was reported in [67] as a dominant failure mechanism, and deeper scratch marks were produced by cracked fillers on worn surfaces. The separation of filler/polymer matrix is understood because of interfacial exhaustion happening in several regions where fillers carry extreme loads [5,61]. Stress transfer and stress concentration at the filler–matrix interface could also lead to significant deformation along the direction of rubbing. Owing to repetitive high stress and strain, specific zones lose their load carrying capability and filler pull-out takes place.

This higher stress also generates cracks in the filler when the highest stress level of fillers ultimately suppresses its strength [5]. Furthermore, the impact that is applied to fillers at protruding areas may also cause filler breakdown when the stress cannot be effectively transmitted to the relatively soft/ductile matrix material. Particularly, this might occur after detachment of the filler/matrix. When the pressure exceeds the critical value, agglomerated nanofillers flatten and no longer play their impingement role effectively. Nevertheless, it was noticed that a change in the speed or dispersion situation of nanofillers might alter the limiting pressure level. In addition, the increase in sliding velocity might be associated with greater contact temperature, with colliding dynamics being applied by protruding areas on the substrate. The rise in contact temperature reduces the matrix stiffness and thus initiates acute stress concentration. Greater contact temperature also reduces the shear strength of the matrix and it could result in an increased wear rate [67]. The worn surface turns out to be smoother at faster sliding speeds, which may reduce the coefficient of friction, as crushed filler agglomerates alter their movement from sliding to rolling [58]. These two aspects can lessen ploughing and cutting influences on pulled-out nanofiller and thus reduce further wear. Throughout the sliding procedure, stress transformation takes place from the matrix to nanofiller in the frictional layer, and consequently the stress concentration on the nanofillers decreases. This morphology is shaped owing to plastic flow

of the surface layer because of the lower stiffness and higher ductility of the polymer matrix [6]. Plastic flow of the PEEK matrix is considerably abridged when nanofillers are integrated into it. Additionally, the improved stiffness of PEEK can diminish the deformation of nanofillers under tension. In general, this is the wear mechanism of nanofiller-incorporated polymer matrix composites, regardless of filler and matrix type and composition.

*2.3. Development High Wear Resistant Polymer Materials with Incorporated Particles*

In literature, a variety of fillers in different forms, such as shape, size, and nature, have been added to the neat polymers with the objective of enhancing their wear resistance [32,33,45,47,48,61–63]. Kurdi et al. [32] reported the effect of nanofiller contents on wear and friction behavior of $TiO_2$-incorporated PEEK polymer, as shown in Figure 4. Up to 5% addition of $TiO_2$ nanoparticles in PEEK reduced the wear rate most effectively, as compared to pure PEEK, although beyond that, a rise in wear rate was noted. This offers an understanding of the critical nanofiller amount in PEEK. This is because of the formation of the effective lubricating TLs in the tribo-contact zone in dry conditions, whereas the contribution of nanofillers to friction is negligible.

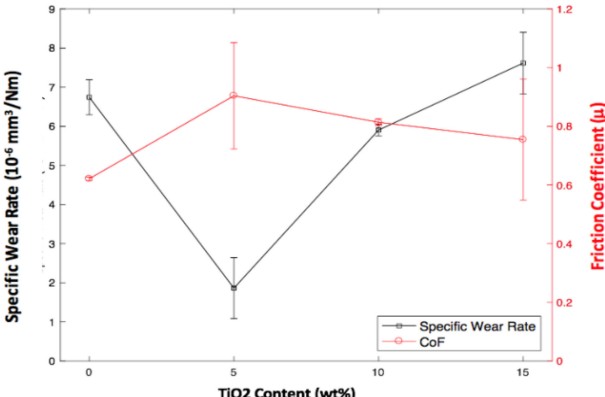

**Figure 4.** Effect of nanofiller content in high-performance polymers (HPPs) on friction and wear [32].

Further, a theory was proposed by Zhang et al. [33], correlating the tribological property of PEEK to its viscoelastic property, the effect of interface temperature, and strain rate of the surface layers (TLs)—all of which are associated with the friction procedure. Beyond the critical level, nanofillers protrude through thin TLs and they act as third bodies to increase the friction coefficient. For wear resistance, the addition of $TiO_2$ nanofiller was useful, as an escalation of the friction coefficient might occur related to the reduction of the specific wear rate, which might be explained by the generation of TLs and will be further discussed in later sections.

However, in reality, some fillers improve wear but deteriorate friction behavior, and trends in the opposite direction have been reported in literature. With the increase of nanofiller content, there is a high possibility of particle agglomeration in the polymer matrix and the effectiveness of nanofiller addition is suspended. A possible solution is using a combination of fillers in the neat polymer matrix instead of a single one. Such combinations, like a blend of fiber- and particle-type fillers, is more effective when compared to a single filler of similar content, as reported by Friedrich et al. [4]. As given in Figure 5, there is about a 300-fold increase of wear resistance of neat epoxy as a result of incorporation of a combination of nanofillers and traditional micro-sized fillers. During the wear process, there is a synergistic effect between short fibers and nanoparticles, which can resist the exfoliation of the surface and accommodate fine wear debris forming effective and durable TLs. As will be presented in later sections, an effective and durable TL not only keeps the coefficient of friction at a manageable scale, but it also reduces the wear rate by altering the wear mechanism to a greater extent. Österle et al. [64] evaluated the outstanding tribological behavior of a polymer composite incorporated with an amalgamation of micron-sized carbon fibers and nanosized silica particles.

Silica-based tribo-films firstly prevent severe oxidational wear, followed by preventing pull-out and rupture of carbon fibers in the composite by offering a cushioning effect. Xie et al. [1] reported that carbon fiber and potassium titanite whiskers (PTW) also function synergistically to boost the wear resistance of hybrid PEEK composites. Furthermore, the carbon fiber transferred the main loads between the contact surfaces and shielded the matrix from additional austere abrasion from the counterpart. Zhang [35,65] reported that adding up of 20 wt. % nanosilica particles boosted the modulus and hardness by 78% and 130%, respectively, as compared to that of pure epoxy. In addition, they also emphasized that the size of the wear debris on TL formation was influenced by the quantity of nanoparticle fillers.

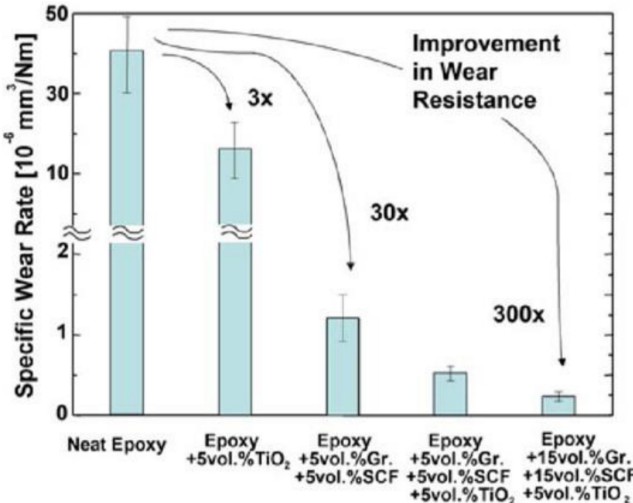

**Figure 5.** Specific wear rate for various degrees of the combination of incorporated particles in neat epoxy (Graphite (Gr), short carbon fibers (SCF)) [4].

Based on the discussions mentioned above, it can be summarized that a variety of fillers are being used in HPPs to enhance their wear resistance. In general, hybrid type fillers, such as a combination of fiber and particles, are more effective than single fillers. Studies also showed that nanometer-sized particle fillers are more effective than micron-sized ones, though their cost-effectiveness is still uncertain.

*2.4. Wear Mechanisms of HPPs in Dry Sliding*

2.4.1. Transfer Layers

The importance of a TF for the tribological performance of polymers has long been realized and widely studied [68–95]. Recently, efforts have been directed to quantitatively characterize TLs and thus to establish more accurate correlations between TLs and the involved wear mechanisms. Chang et al. [54] studied TL formation for different polymer-based hybrid composites by applying nanoindentation with in-situ AFM examination. Based on wear results and indentation analysis, it was noted that the hybrid nanocomposites that were incorporated with both nanoparticles and conventional tribo-fillers were useful for forming long-lasting TLs, particularly in severe sliding circumstances, and subsequently reduced the friction coefficient and wear rate. A synergistic action between nanoparticles and TLs is presented that might be the key mechanism for the enhanced wear characteristics of polymeric hybrid nanocomposites. Generally, a reduction of the coefficient of friction and wear rate takes place with increased TL thickness and even coverage. In most tribological uses, the TL controls load transmission and the wear mechanism. Therefore, trustworthy material data for TLs is advantageous [6,31]. TL behavior is often treated as a signature for different combinations of polymer/incorporated particles. Some blends affect the progression of TLs, whereas others do not, and consequently increase wear rather than reducing it [31]. It was stated that when polymeric composites are incorporated with conventional fillers, the composites were more useful for forming

durable TLs on a sliding steel surface, regarding their tribological characteristics [7]. It was also proposed that the brittle–ductile transition of polymers plays a vital role in controlling the development of TLs when temperature increases [14].

The effect of nanofillers on enhanced TFL generation is generally considered to be more effective than that of micro/macrofiller, as established in the literature [15,32–34,47,48]. This was commonly attributed to the higher surface-to-volume ratio of nanofillers, which provides the more interfacial areas at the same volume fractions compared to microfiller-incorporated blends. Also, wear debris is comparatively bigger in size and shape for micro/macrofiller-incorporated composites, which does not favor the formation of continuous, uniform TLs. In fact, TLs form as compacted wear remains and are greatly associated with the use of nanofiller-incorporated blends. TLs that are generated in the polymer-on-metal sliding process are dissimilar to traditional lubrication films. Gao et al. [19,46] carried out a detailed investigation on the nature of TLs that form on steel during sliding against PEEK composites with the help of focused ion beam and scanning electron microscopes (FIB-SEM) and Transmission electron microscopy (TEM), as shown in Figure 6. It was evident that the accumulation of TLs on steel is uneven in nature. Moreover, the diffraction pattern of the selected area contains both hollow rings and dots, which represent the amorphous and crystalline nature of the TLs.

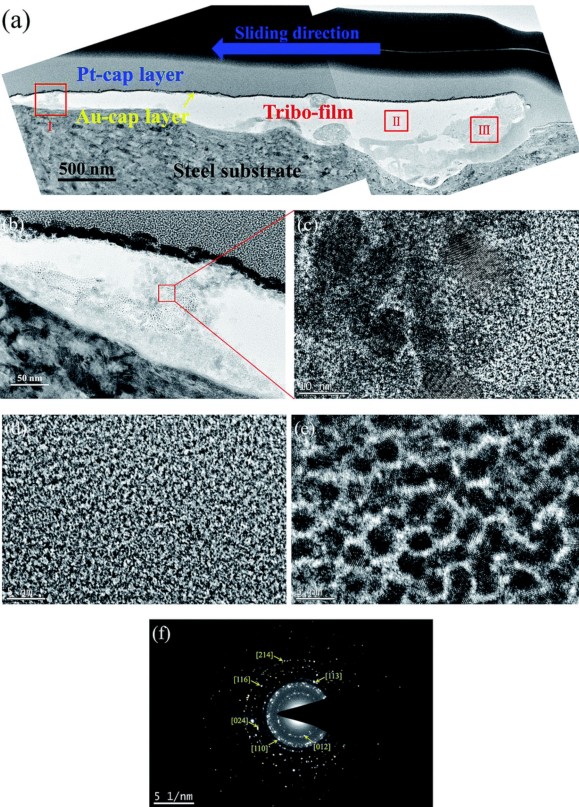

**Figure 6.** TEM micrographs of transfer layers (TLs) on the cross-section of steel counterpart after sliding on PEEK/10FeOOH at 100 N normal load: (**a**–**e**) bright field TEM pictures; and, (**f**) selected area diffraction pattern on TLs [19].

The thickness of TLs can be also determined by nanoindentation based on a relatively simple model, as described by Bahadur et al. [95,96] and Chang et al. [37], where composite hardness, $H_c$ of the film/substrate arrangement while indenting through the film onto the substrate can be defined from the portion of contact areas [37,95,96]:

$$H_c = \frac{A_f}{A_t}H_f + \frac{A_s}{A_t}H_s \qquad (2)$$

where $A_t$ is total projected contact area of indentation, $A_s$ and $A_f$ are the projected contact areas of the substrate and film, respectively, and $H_f$ and $H_s$ are intrinsic harnesses of film and substrate, respectively. Since $A_t = A_f + A_s$, Equation (2) can be rearranged as:

$$H_c = \frac{(A_t - A_s)}{A_t} H_f + \frac{A_s}{A_t} H_s = H_f + \frac{A_s}{A_t}(H_s - H_f) \tag{3}$$

$$\frac{\left(H_c - H_f\right)}{\left(H_s - H_f\right)} = \frac{A_s}{A_t} \tag{4}$$

For sharper indenters, for example, the Berkovich, indenter contact area is proportionate to the square of indention depth, i.e., $A = k \cdot h^2$ and thus, Equation (4) can be rearranged as:

$$h_f = h_t - h_s = h_t \left(1 - \frac{h_s}{h_t}\right) = h_t \left(1 - \sqrt{\frac{\left(H_c - H_f\right)}{\left(H_s - H_f\right)}}\right) \tag{5}$$

where $h_t$ is total indentation depth, $h_f$ is TLs thickness, and $h_s$ is indentation depth in the substrate. The hardness of TLs, $H_f$, was considered to be similar to the neat polymer matrix hardness. In addition, the variation of hardness between the steel counterpart and neat polymers varies largely. Therefore, the minor deviances in $H_f$ due to the presence of harder minor pieces in soft TL might be overlooked. It is noteworthy that a strong variation in hardness along the wear trails was commonly observed, which implies the nonuniform allocation of TLs along the wear track. On the other hand, a thicker TL corresponds to lower magnitudes of $H_c$. A soft TL is usually pushed out from the indentation region, generating a "pile-up" of material nearby the indentation due to the confinement of plastic flow of polymer TLs by the hard substrate beneath [37]. With the estimation of film thickness, the transferal film efficiency factor (λ) can be defined by applying Equation (4) [37]:

$$\lambda = \frac{t}{R_a} \tag{6}$$

where $t = (h_f)$ is the average TL thickness established by nanoindentation and $R_a$ is the surface roughness of the steel surface. Further reading on the equation's development and the concept of TLs can be found in the literature [37].

With the transferal film efficiency factor, the tribo-effects TLs can be further described, showing the similarity to well-established Stribeck curvatures [39]. Santer et al. [97] extended this in wear systems as a function of relevant factors [97]. Lately, Krudi et al. [32] also reported a similar effect of TLs on the coefficient of friction of various $TiO_2$-incorporated PEEK composites, as shown in Figure 7. TLs share the similar physical meaning of those conventionally applied variables [32], i.e., a higher magnitude of the parameter points out that the characteristics of TLs further control the frictional property of the sliding pairs. Despite the desirable wear performance that is achieved by the presence of TLs, it was also observed that the spreading of TLs was uneven, as mentioned earlier. Therefore, during nanoindentation, in order to find the thickness of TLs, a sufficient number of tests should be performed along the wear track for the statistical purpose. Though the magnitudes of hardness might be varied broadly between the hardness of virgin steel counterpart and that of the polymer matrix, the average magnitude of hardness exhibited decent replicability with an adequate number of indentation tests (e.g. 100 times) over certain length (e.g. 1 mm) and the curves displayed comparable trends. Since the film was discontinuous, such average magnitudes would not be assumed as the general characteristic of a comparable film. The 'mean thickness' is considered as a quantifiable guide of the total amount of TLs covered on the counterpart. The unevenness of TLs on substrate restricts the accuracy of the calculated thickness. Additional shortcomings arise due to other parameters, for example, indenter tip sharpness, the surface finish of sample, pile-up, etc., which induce uncertainties in defining the actual contact

area during nanoindentations. Nonetheless, the investigation offers a practical technique to define and compare the TLs generated on a range of materials in various sliding circumstances. Regardless of '*pv*' factor, the common tendency was that the composites incorporated with nanoparticles experienced thicker TLs, as compared to those without nanofillers. The literature indicated that TLs generated by neat polymers has the tendency to form thicker TLs at faster speeds but thinner ones at larger pressures [8,60]. As pointed by Bahadur et al. [95,96], wear is also dependent on the cohesion strength of TLs on counterpart. Friedrich et al. [34], however, emphasized more on the thickness of TLs. According to SEM observation, it was proposed that when the TL is thinner than substrate's roughness the effect of TLs is less dominant, and breakage of TLs is more likely. In the case of the higher surface roughness of mating pairs, more debris is required to fill up the valleys, associated with a high abrasive wear in running-stage. After filling the valleys with wear debris and forming semi-continuous TLs, however, the detachment of large chunk of TLs could still occur, leading to a high-wear process, even after the running-in stage.

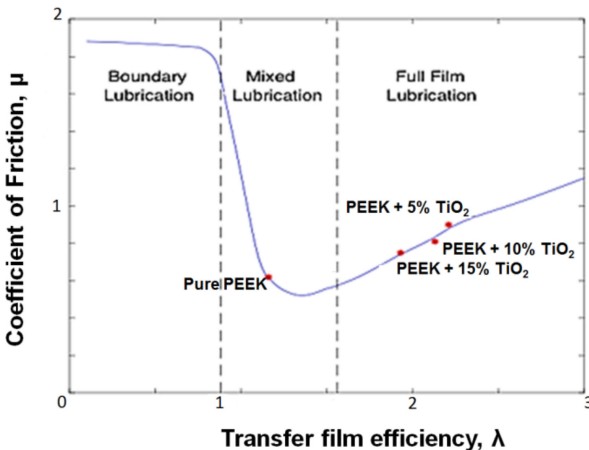

**Figure 7.** Stribeck type curve under dry condition as a function of the coefficient of friction and transferal film efficiency factor [32].

### 2.4.2. Wear Mechanism in the "Steady State" of Sliding Wear

As aforementioned, the TLs would be gradually developed on the metallic counterpart, during the initial running-in phase. Subsequently, the overall friction and wear of the tribo-system become rather steady. In the steady state, with the presence of the TLs, the mode of tribo-contact changes from the initial hard (metal)—on—soft (polymer) to one or more of the subsequent contact couples, depending on the thickness, distribution, and components of TLs, namely, (a) hard-on-hard (rigid fillers against asperities of steel surface), (b) hard-on-soft (rigid fillers against polymeric TLs), and (c) soft-on-soft (polymer against polymeric TLs). Thus, the contact mode relies on the spreading and quantity of TLs in actual contact area.

In respect of the values of λ, three different wear- and friction-regimes might be identified, as shown in Figure 8 by Chang et al. [37] who have investigated the tribological aspect of HPPs-based composites filled with various fillers. When the magnitude of λ is comparatively low (< 0.2), the contact is considered as insufficiently lubricated, as there is not enough TLs to completely cover the steel counter-face. In this situation, wear behavior greatly depends on sliding circumstances, such as "*pv*" factors, as described in Section 2.1. When the product of "*pv*" is small, it is likely to shield the surface efficiently, even with thin TLs, because of a small actual contact area. Polymer composites might attain decent wear characteristics with a relatively smoother worn surface. However, under greater "*pv*" values, the actual contact area rises, and additional fillers, such as fibers, are exposed to steel counter surface with insufficient shielding from TLs. The contact type would be in the form of "hard-on-hard", resulting in a high coefficient of friction. In addition, fillers are more probable to break down and pull out rapidly. The wear procedure could be further intensified by the thermal assisted mechanical

failure of polymer matrix, particularly in the interfacial area. Therefore, polymer composites risk acute wear loss at severing sliding circumstances with thin TLs. This has somewhat resembled boundary lubrication situations. With rising magnitudes of λ, TLs can be considered as adequately lubricated layers where they are capable of shielding most of the fillers from direct contact with steel counter-face, for both high and low "*pv*" factors. Wear behavior is determined by a combination of two contact modes: (a) contact of the polymer against TLs (soft-on-soft) and (b) contact of hard incorporated fillers against TLs (hard-on-soft). Due to the shielding of TLs, severe fillers pull-out might be evaded, even at high "*pv*" conditions. Nevertheless, if the quantity of TLs is very high, this might be also related to the high wear loss. In this case, TLs can break down in the form of the large chuck of wear debris as relatively thick TLs could act as thermal insulators on steel surfaces. Hence, localized contact temperature in that thick TLs area might be high and may cause strong adhesion between polymers and TLs. On the other hand, high contact temperature may decrease the viscosity of polymeric TLs, which sequentially results in a small coefficient of friction. In view of that, there is no such a direct correlation between the coefficient of friction and wear of material, even when taking consideration of the role of TLs. However, it should be stated that, at greater quantities of λ, the tribological behavior of sliding system is mostly controlled by the characteristics of TLs. Therefore, under such circumstances, the wear behavior of HPPs displays less dependency on sliding circumstances. Though the TLs showed the resembling lubrication influence to that of conventional lubrication films, there are more complex interactions between tribological characteristics of TLs and polymers. One example is that TLs are formed directly by the wear debris of sliding pairs while the traditional lubrication films are normally formed by external lubricants. Additionally, the rise of temperature in solid contact due to sliding might significantly affect the characteristics of TLs, which, in turn, influences the tribological behavior of the sliding pairs [98]. For example, a small coefficient of friction was attained for PA66 matrix composite with no nanoparticles, at the expense of surface melting due to high contact temperature. As the thermal conductivity of steel counter surface (around 58 $Wm^{-1} K^{-1}$ [99]) is considerably greater than that of polymers (about 0.25 $Wm^{-1} K^{-1}$ for PA66 [99]), a thicker TL may also work as a thermal shield, as mentioned before. In case the thickness of TLs is greater than the surface roughness of the steel counter surface, the effect became more significant. The pulped TLs might contribute to a small coefficient of friction due to the lubrication effect. However, the melted polymer in the contact region cannot effectively carry the load and sometime failed to protect filler pull-out, associated with high wear loss. Henceforth, tribological effects of TLs always require being considered cautiously, on the basis of a good knowledge of the wear/contact mechanisms of the system in the given sliding condition.

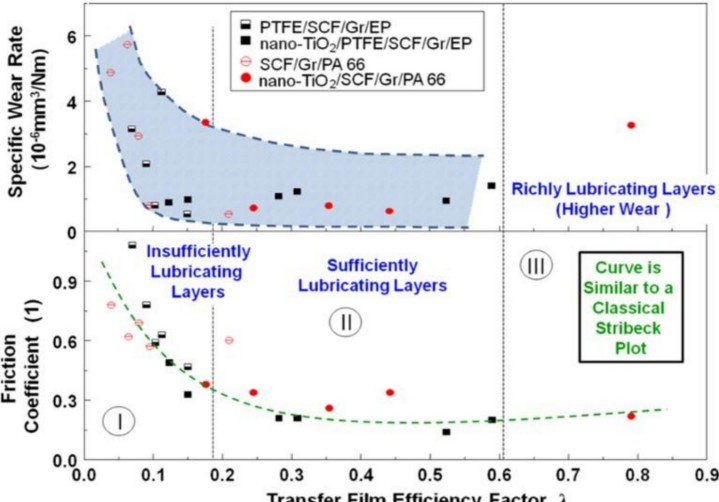

**Figure 8.** Evolution of the coefficient of friction and specific wear rate based on the transferal film efficiency factor that rises to different boundary regimes [37].

The experimental data have shown that more durable TLs could be expected by adding more than one type of tribo-fillers in neat polymers. According to Kurdi et al. [32], the use of sole TiO$_2$ nanoparticles is not as effective as the combination of such nanoparticles with short fibers, solid lubrications, etc. [27,36]. The outcomes suggested that the lower friction characteristic of nanofiller-incorporated composites resulted from the mechanical interactions among supplementary nanofillers, polymeric nature of TLs, and contact surfaces [29,36]. Nanofillers can contribute to a lower coefficient of friction, because of their 'spacer' effect, as well as its capability of rolling [36]. These effects of nanofillers would also favor the formation of durable TLs. The existence of nanofillers may also reduce the adhesion between TLs and polymers by decreasing the real contact area. At the same time, nanofillers could be entrenched into soft TLs, instead of being pushed away from the compressed surfaces. Thus, TLs partly cover the nanofillers and minimize their abrasion effect. Such a synergistic effect between TLs and nanofillers is useful for an improved wear characteristic of polymeric composites, particularly under acute sliding circumstances.

As discussed in the literature [9], at the microscopic level the contact between mating surfaces during sliding can be resolved as a summation of a number of asperities contact. The nature of continuous/semi-continuous transfer film can be explained in view of such asperities contact, as schematically shown in Figure 9. As the pin (test coupon) travels on the hard counter surface, the front edge of asperities cut the bulk material from the pin, and the removed materials can be accumulated by front flanks. In the course of that, the tail edges of the asperities keep exposed, as they do not contact the bulk material. This phenomenon gives rise to the characteristic, 'rippler' like appearance of the counter surfaces, as stated in the literature [9]. As the process continues, the space between asperities get filled up with wear debris, forming the TL. With the development of a stable TL, the wear process enters the "steady-state", where both the coefficient of friction and wear rate dropped, as compared to the "running-in" phase. From this point onward, the governing wear mechanism, as well as the wear rate of the specimen is dictated by the retention ability of the TL. Nevertheless, it is worth indicating that additional tribo-fillers in the neat polymers do not always yield positive effect on TL formation, as presented by Bahadur et al. [15]. The beneficial effects of tribo-fillers are normally associated with the improved adhesion of the transfer layer on the counter surface, depending on fillers' compatibility and chemical nature.

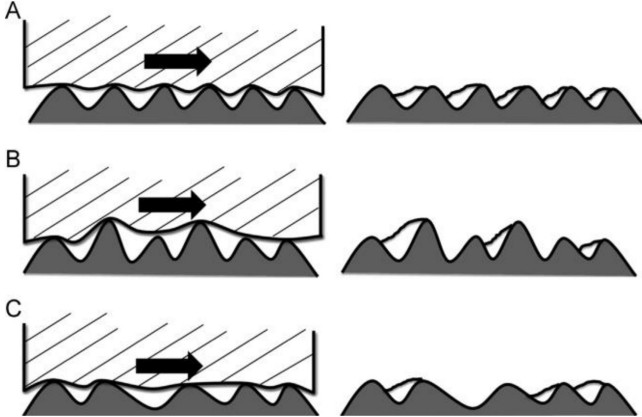

**Figure 9.** Schematic of asperity role on transfer layer formation: (**a**) well-distributed height and spacing of asperity, (**b**) relatively taller peaks of asperity, and (**c**) asperities with wider gaps [9].

## 3. Effects of External Environmental Conditions

The development of HPPs aims to bring new solutions for mechanical engineering applications under more complicated, harsh environmental conditions e.g., with higher service temperature or corrosive liquids. To date, however, little effort has been made to understand wear behavior under those conditions. In this paper, the effects of external conditions, such as high temperatures, lubricants, and vibration, will be particular reviewed.

### 3.1. High Temperatures

Just a few decades ago or so, as compared to thermoplastics, thermosetting polymers were normally favored for engineering applications, owing to their desirable properties, such as heat resistance, mechanical durability, and dimensionally stability. Recently, new thermoplastic polymers have been successfully synthesized, having the increased strength by a factor of 2–2.5 and the high operating temperature up to 100–150 °C, together with the enhanced water and chemical resistance and reduced flammability [96]. Progress in this field was so significant that the new materials are termed as "super thermoplastics" or "super constructional thermoplastic polymers". They can be applied in extreme conditions as polymer matrixes for composites, instead of traditional thermosetting phenolic and epoxy resins [3,38]. HPPs, namely polyetheretherketone (PEEK), polyphenylene sulphide (PPS), polyetherimide (PEI), poly ether sulfone (PES), and thermoplastic polyimide (TPI) are attributed to the group of heat resistant super thermoplastics for high-temperature applications [21]. Their molecular structures contain hard and heat resistant fragments along with simple ester, sulphide, amide, and ester groups. Ether and sulphide groups act as joints that provide chain flexibility without reducing heat resistance. Polymers of this group are characterized by a strong intermolecular interaction. This factor, coupled with "joints" provides a high modulus of elasticity with considerable tensile elongation and high shock resistance. Due to this fact, such super thermoplastics are distinguished by high strength characteristics, which are 5–10 times higher, on average, than those of epoxy based thermosetting resins. Modulus of elasticity of the heat resistant super thermoplastics is as high as 2.5–4 GPa, with tensile strength in the range of 70–130 MPa. Most of such materials retain good physical and mechanical properties over a wide temperature range. The heat resistance of such polymers is not only due to its chemical structure, but also on phase state-particularly the ratio of crystalline and amorphous phase together with filler dispersity [21,31]. For thermoplastics, having partially crystalline structure (PPS, PEEK), heat resistance is determined by the melting point of crystalline phase, which is higher than the glass transition temperature by 150–180 °C. As fillers initiate the formation of the crystalline phase, heat resistance of PPS and PEEK-based composites can be higher by 100–160 °C as compared to neat polymers [73]. As mentioned in the introduction section, HPPs were developed with the intention to use them in harsh conditions, especially where service temperature is high.

In general, the effect of temperature on tribological performance of polymers at the certain temperature can be understood while considering two aspects of the tribo-system: (a) contact conditions and (b) the micro-structure of polymeric specimens, which are sensitive to the contact temperature. Normally, temperature rise tends to defoliate the strength of polymers. As a result, the material became readily flow able and thus gave rise to the stick-slip condition. In the stick procedure, there is no relative removal between counter-body and polymer surface because of a strong adhesive force. Therefore, the sliding surface exhibits a great tangent deformation and material gathers in front of the sliding counterpart. When the tangential stress is less than the critical stress the contact region between mating surfaces grows with the time. When the exerted stress on polymer surface surpasses the critical stress [90–93], the slip process occurs until the contact stress decreases below that critical level. Additionally, material pile-up could result in the strain hardening effect, because of the tangles of long molecules [36]. As a result, the slip stage can actually be close to the stack region of polymers. In addition to that, as confirmed by DMA analysis, temperature rise generally decreases the storage modulus of materials [93]. However, a rapid decrease in Young's modulus takes place simultaneously, especially during the transformation from glassy state to viscoelastic state. This would lead to an increased contact area with higher friction coefficient. Besides, the increased hysteresis loss at higher temperature might also be an important factor, accounting for the increased friction coefficient [94]. Samyn et al. [56] demonstrated that global bulk temperature is vital for changeovers of friction, while the wear mechanism is more controlled by the localized temperature. The polymer material might be transported to the mating surface gradually. Lastly, the gathered material on the mating surface detaches in the form of large pieces [56]. As confirmed by SEM analysis of wear tracks,

large sheet-like debris was sometimes detected on (or near) wear track, which could be defined as a "transfer" mode [27].

Up to now, the sliding wear behavior of various HPPs have been studied at elevated temperatures. Friedrich et al. [2–4] reported that poly (p-phenylene) (PPP) is a suitable candidate for numerous engineering purposes as far as the temperature remains below 140 °C. However, sliding wear performance was not as great as that found on neat PEEK. Chang et al. [52] studied the influence of nanofiller addition on tribological properties of PEI and PEEK composites at both room and high temperate. They conclude that, though the addition of sub-micron particles does not seem to improve wear resistance at room temperate, the remarkably enhanced wear resistance at elevated temperatures was noticed. Chang et al. [54] further investigated the wear characteristics of PBI, as well as PEEK at elevated and room temperatures. According to their report, PEEK could achieve higher wear resistance than PBI at the higher temperature, in spite of the deterioration of its mechanical properties.

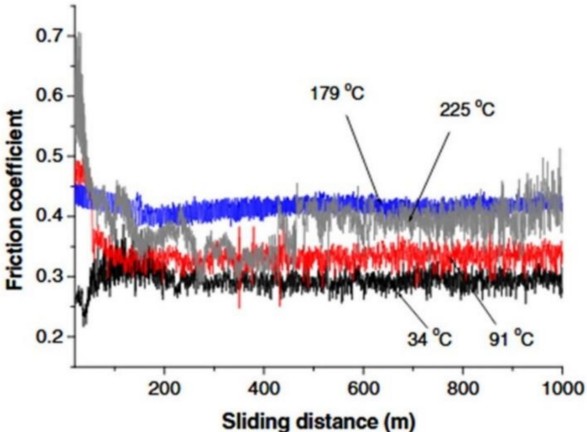

**Figure 10.** The coefficient of friction of PEEK with the increase of sliding distance at different temperatures [55].

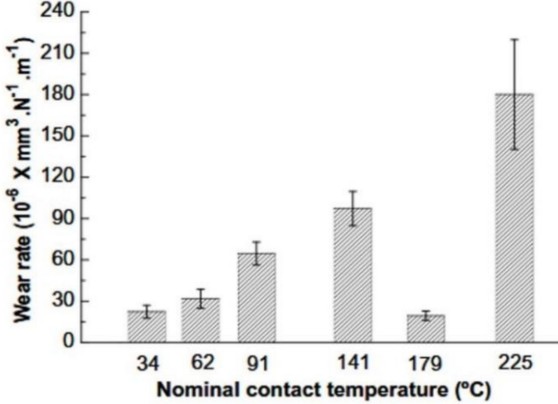

**Figure 11.** Wear rate of PEEK at different nominal contact temperatures during dry sliding against 100Cr6 steel ball [55].

More detailed studies on the temperature dependent tribological behavior of pure PEEK were carried out by Zhang et al. [55, 67]. As shown in Figure 10 [55], friction coefficient initially increased with temperature. However, there was a transition in friction coefficient when the temperature increased from 179 °C to 225 °C. This transition is also evident in the evolution of wear rate, as depicted in Figure 11, [55]. To understand such a transition, Zhang et al. [35,60,65] further investigated the influence of temperature on structural changes of polymers before and after sliding tests. Based on XRD and DSC analysis, it was concluded that external temperature could affect the amorphous/crystalline transition of PEEK when the temperature is comparable to materials' glass transition temperature ($T_g$). The thermal properties of different polymers, such as glass-transition temperature ($T_g$), melting

temperature ($T_m$), and heat deflection temperature ($HDT$), together with their respective coefficients of friction ($\mu$) are summarized in Table 1. It is worth noting that PEEK has high melting ($T_m$) and glass transition ($T_g$) temperatures of 343 and 143 °C, respectively [1–3]. Before crystallization, both the coefficient of friction and wear rate increase with temperature, due to the increased contact area and adhesion. Once crystallization/semi-crystallization of the material took place, the tangential plastic deformation of PEEK surface is relieved and the space among intermittent material piles is decreased. Moreover, the crystallization/semi-crystallization of PEEK causes an increase in storage modulus of the material, contributing to a smaller contact area. The limited lamellae structure in the amorphous phase efficiently restricts the tangent deformation of the surface layer. In this case, stress concentration likely progresses whereas shear force is in action. Thus, due to the repeated stress, fatigue comes into consideration and it contributes significantly to the wear process. Overall, the friction and wear decrease, as compared to the severe adhesive wear at relatively lower temperatures. This explains that, up to the certain temperature, the wear resistance of PEEK deceases with temperature and then increases. A similar temperature effect on friction and wear behavior of HPPs was also reported by Kurdi at al [98], as shown in Figure 12. The results also confirm there is not such a simple and direct correlation between wear rate and friction in terms of environment temperature or internal properties.

**Table 1.** Thermal properties of different polymers as available in literature.

| Polymers | Coefficient of Friction ($\mu$) | Glass-Transition Temperature ($T_g$) | Melting Temperature ($T_m$) | Heat Deflection Temperature ($HDT$) |
| --- | --- | --- | --- | --- |
| PET | 0.33 [20] | 60 [74] | 252 [74] | 62 [74] |
| PSU | 0.37 [2] | 85 [76] | - | 174 [76] |
| PES | 0.62 [76] | 225 [76] | 200 [76] | 203 [76] |
| PVDF | 0.24 [56] | - | 170 [76] | 260 [76] |
| PEI | 0.1 [58] | 230 [76] | 190 [76] | 210 [76] |
| PPS | 0.43 [47] | 83 [74] | 285 [100] | 108 [76] |
| PEEK | 0.40 [1–3] | 143 [1–3] | 343 [1–3] | 152 [76] |
| PPP | 0.87 [98] | 150 [89] | 340 [3] | - |
| PBI | 0.67 [98] | 400 [54] | - | 427 [94] |
| PTFE | 0.1 [2] | 27 [76] | 325 [76] | 90 [76] |
| PI | 0.48 [32] | 320 [67] | 385 [76] | 238 [76] |
| EP | 0.53 [35] | 132.5 [57] | - | - |
| PAR | - | 190 [25] | - | 174 [76] |
| PC | 0.45–0.55 [76] | 145–148 [67] | 260–270 [67] | 129 [76] |
| PET | 0.141–0.245 [76] | 80 [76] | 250 [76] | 66 [59] |
| PEK | 0.29 [76] | 165 [92] | - | >316 [22] |
| TPI | 0.43 [32] | 250 [78] | - | 332 [23] |

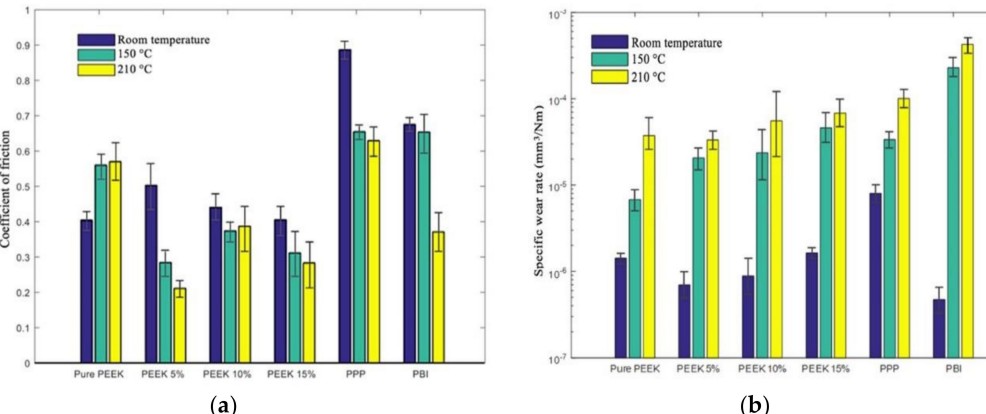

(**a**)          (**b**)

**Figure 12.** Influence of environment temperature on (**a**) the coefficient of friction and (**b**) specific wear rate of HPPs [98].

### 3.2. Lubricated Conditions on Friction and Wear

A number of publications are available in literature regarding the tribological behavior of HPPs sliding under both dry and wet conditions [5–8,14–21,71]. Polymers are widely used under dry sliding conditions, thanks to their self-lubricating behavior. However, water (or another liquid medium) is sometime inevitably involved in tribological applications for HPPs. In this case, water might still perform as a cooling agent and thus, friction heating is effectively subdued. In fact, the temperature rise in an aqueous environment under different loadings is normally less than 10 °C. As generally expected, the introduction of a liquid lubricant in tribo-contact will decrease the coefficient of friction and wear. This is not always true for polymers. In practice, some HPPs even exhibit higher wear loss in certain lubricant mediums.

Gao et al. [46] investigated the wear and friction behavior of epoxy polymer composites using water as a lubricant. Based on their findings, it was concluded that the incorporation of short glass fibre (SGF) or short carbon fibre (SCF) in epoxy polymer is not beneficial, whereas graphite particles could reduce the coefficient of friction without having any positive impact on wear rate. Yamamoto et al. [73] studied the effect of water as the lubricating medium on the tribological behavior of PEEK and PSS in both similar and dis-similar tribo-contact. Both PPS and PEEK are recognized as high hydrolysis as well as chemical resistant polymers [74,75]. As shown in Figure 13, the effect of water in both cases is evident, as the coefficient of friction decreases in all cases. With the presence of water film, direct rubbing between tribo-contact is avoided to some extent, leading to a lower coefficient of friction.

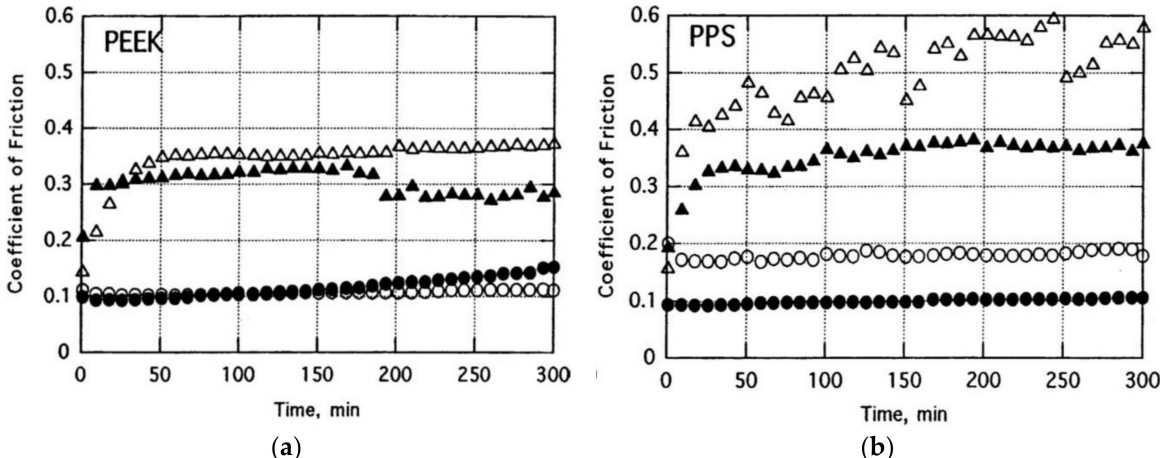

**Figure 13.** Effect of water as a lubricant on the coefficient of friction of (**a**) PEEK and (**b**) PPS under different contact conditions: polymer/polymer (△) dry; (▲) in water and steel/polymer (○) dry; (●) in water [73].

Though water results in a positive effect on friction reduction, the scenario is completely opposite in the case of wear, as shown in Figure 14. It was noted that the hardness of PEEK decreased as a result of sliding in water, though it did not happen only by immersing the polymeric specimens in water. This means that the water absorption in PEEK only took place in wear process under certain external loads. With respect to that, it is suggested that HPPs, such as PEEK, are most suitable for applications where the external liquid lubricant is not permissible. The authors [73] have also investigated the tribo-effects of lubrication as "bearing modulus ($\varphi$)" and claimed that glass fibers incorporated PEEK composite displayed meager wear performance in water lubricating conditions, as compared to that of carbon fiber incorporated PEEK composite. In contrast, neat PPS exhibits better wear and friction behavior than that of PEEK, whereas the incorporation of carbon or glass fibers further decrease the wear rate with an increased friction coefficient. Thus, not only the type of lubricants but also the nature of fillers in HPPs composite, as well as their interactions contribute towards the overall wear and friction of the sliding system. In particular, carbon fiber incorporated PPS showed greater wear resistance in water lubricant than glass fiber incorporated PEEK with the

specific wear rate of $0.2 \times 10^{-6}$ mm$^3$/Nm and $1 \times 10^{-6}$ mm$^3$/Nm, respectively. For all PEEK-based composites, the specific wear rate at sliding speed beyond 2 m/s was relatively low, that is about $10^{-8}$ mm$^3$/Nm as compared to $10^{-6}$ mm$^3$/Nm at higher sliding speed. The sliding surfaces of PEEK composites experienced less damages when compared to that of neat PEEK. As evident from Figure 14, in both lubricated and unlubricated conditions, the wear rate of PEEK is a couple of folds higher than that of PPS, under similar sliding conditions. The associated wear mechanisms were attributed to: (i) TLs formation, (ii) the adsorption of water molecule, and (iii) the chemical bonding of transfer layer onto the counter surface. In unlubricated conditions, the pronounced TLs were noticed in the case of PPS as compared to PEEK, which was further tarnished in water-lubricated conditions. The better ability to form TLs by PPS is due to a more effective chemical bonding between the TL and steel counter surface. In contrast, the high wear rate of PEEK is due to lowering its mechanical strength/hardness at the sliding area under normal/tangential forces as a carbonyl group (–CO–) of PEEK molecular structure was interrupted by water molecules and contributing towards bulk plasticization [76].

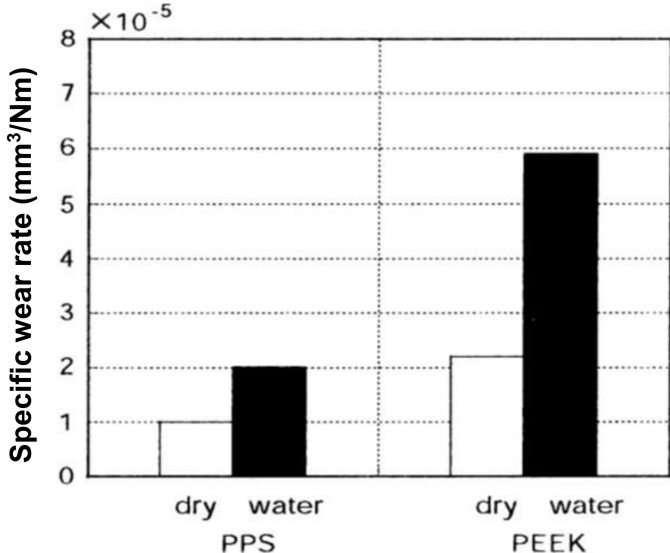

**Figure 14.** Effect of water lubricant on the wear rate of HPPs [73].

The roughening of sliding surfaces as well as the transmission of materials to steel counterpart were repressed by PEEK incorporated with fibers. In view of boundary lubricating circumstance, the reduction in harshness of contact between sliding surfaces reduces the specific wear rate. This was more pronounced at sliding velocities of 2 and 4 m/s, where the lubrication regime appeared to be hydrodynamic one, as the value of subsequent bearing modulus ($\varphi$) ranges from $9 \times 10^{-8}$ to $2 \times 10^{-7}$ [73]. As long as the hydrodynamic or mixed lubrication circumstances could be maintained, or the degree of direct contact was maintained as insignificant, glass or carbon fibers were useful in reducing the surface damage of PEEK composites. Neat PPS itself displayed better wear behavior as compared to neat PEEK and the addition of incorporated particles did not offer any distinct improvement. PPS composites displayed greater specific wear rate than PEEK composites in the hydrodynamic regime at the velocity beyond 2 m/s with the equivalent $\varphi$ value being greater than $9 \times 10^{-8}$. Thus, with hydrodynamic lubrication, PEEK composites might be more promising than PPS composites. Under the lubricated condition, the contact stress can be compatibly high and complexities might rise due to the generation of hydrodynamic films. With the purpose of minimizing hydrodynamic influence, it is desirable to select the input variables so that the "wear scar diameter" [77–79] is less than the corresponding critical value all of the time. The capability of the liquid lubricant to be engrossed by the polymer is controlled by the values of solubility variables of the fluid ($\delta S$) and polymer ($\delta P$). The critical stress is smaller when $\delta S$ is closer to $\delta P$. In this case,

the propensity to crack is higher than crazing, as reported for the case of polyamide 66 ($\delta P = 28$ $(\text{M·J·m}^{-3})^{1/2}$) in water ($\delta S = 48.5$ $(\text{M·J·m}^{-3})^{1/2}$).

More recently, Kurdi et al. [32] also investigated the effect of water as lubricant on PEEK composite filled with various TiO$_2$ contents, as shown in Figure 15. It is obvious that, as the friction coefficient decreases in the presence of water, the wear rate increases to a considerable extent. They also pointed out that the degradation of polymer structure as a result of water molecule absorption, easing the exfoliation of polymer, is the main reason for such a high wear rate as compared to dry sliding condition. The similar tribological behavior of different polymers and polymer composites under water lubricated condition was also reported by other researches. For example, Friedrich et al. [80] studied similar behavior on PET, PI, PEEK, and their composites, Wang et al. [81] on PTFE composites [82], Gao et al. [19,46] on EP and PEEK composite and Golchin et al. [83] on PPS composites. Another aspect of water lubricated condition is the inhabitation of TLs formation on the counterpart. Water runs through the surfaces of polymer composite and the mating component, taking away wear debris. This cleaning process exposes the virgin surface of the polymer to counterpart uninterruptedly. This might explain the result that there is sometime no eminent running-in stage in friction curves with water [84]. Thus, TLs, which is generally noted in dry sliding, fail to generate in the presence of lubricants. Furthermore, the water may cause the chemical corrosion of steel counterpart and thus make the surface rougher as compared to its initial state. The experiments showed that the immersion of steel in water improves its wettability as well as accelerates wear of polymer [85,86]. Besides, the interfacial amalgamation between the matrix and incorporated particles was deteriorated in water, especially under higher load. The deteriorated interfaces might perform as crack nucleators and accelerate the spread of fatigue cracks at sub-surface. The cyclic stresses that were generated in the contact area of the polymer specimen also caused the peeling off of the surface layer. The exfoliation on worn surface suggested that fatigue delamination took place in wear process. Therefore, a huge quantity of sheet like exfoliations gathered on the worn surface, which subsequently caused a severe abrasive effect. Actually, both friction and contact temperature were significantly raised under pressure up to l4 MPa, accelerating the crack propagation of the matrix, particularly in the interfacial area.

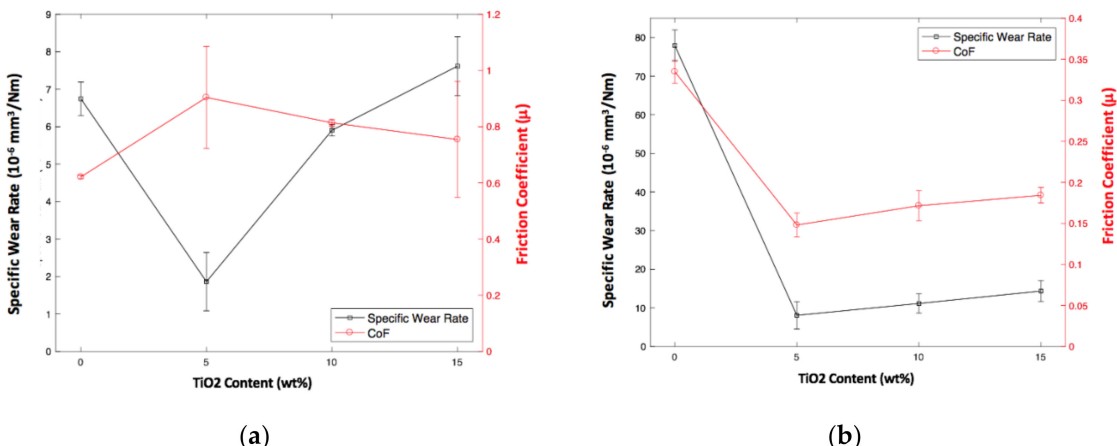

(**a**) (**b**)

**Figure 15.** Effect of water as a lubricant on wear and friction of various TiO$_2$ content PEEK composite (**a**) under dry and (**b**) water lubricated conditions [32].

In addition to the water lubricated condition, Zhang et al. [87] studied the tribological behavior of EP composite under diesel lubricated conditions, as shown in Figure 16. Unlike the results from the water lubricated condition, the wear rate did not increase in diesel lubricant condition, as diesel was not absorbed by PPS composite and maintained the boundary lubrication condition in the form of a thin fluid film. Such a continuous lubricating film efficiently transmits the load and keep both friction coefficient as well as wear in low scale as compared to dry conditions.

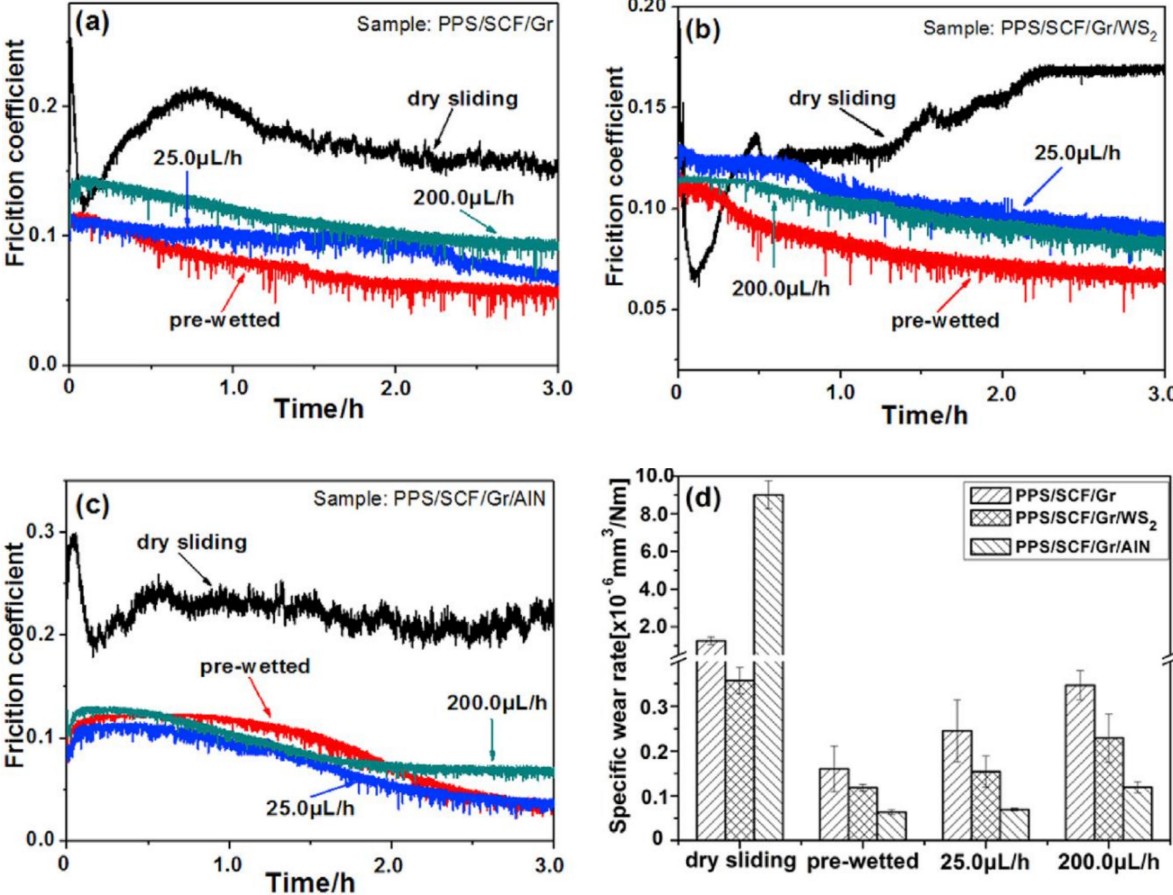

**Figure 16.** Effects of diesel as a lubricant on friction coefficient evolutions of (**a**) PPS/SCF/Gr; (**b**) PPS/SCF/Gr/WS₂; (**c**) PPS/SCF/Gr/AIN, and (**d**) their corresponding wear rates [84].

In view of the above-mentioned discussions, the common lubrication mechanism of HPPs is the formation of TLs, which for most of the cases resemble asa "mixed lubrication" effect. In addition to that, the mechanical and chemical interactions between the polymer, steel counterpart and the incorporated fillers play a vital role for the formation/retention of transfer layer during sliding. For instance, the nature and chemical affinity of the fillers used in the polymer composite may greatly affect the overall wear rate. As reported by Zhao et al. [88], PPS with $Ag_2S$ showed a better wear rate when compared to other fillers during rubbing against steel counter surface. During the sliding process, Fe from the counter surface reacts with S to form FeS and $FeSO_4$, due to higher chemical affinity. Formation of these compounds results in a strong chemical bonding between steel surface and the formed TL, as confirmed by XPS investigation. The durable TL then acted as a stable lubricating like film and effectively reduced the wear rate. This was quite evident, as the neat PPS showed a much higher wear rate under the similar experimental condition, due to the absence of the effective TL.

### 3.3. Impact of Vibration

Vibration can commonly occur for the contacted moving parts in tribological studies. Nevertheless, the effect of vibration on friction and wear is little reported in literature. The general understanding of the impact of vibration is that, under certain frequencies and amplitudes, there could be a physical gap in tribo-contact with an increase in normal load, for a fraction of time. Such a repeated action affects the overall friction and wear behavior. The effect of a perpendicular vibration on friction force is considerable, depending on rubbing duration, amplitude, and vibration frequency. Research shows that, time to obtain steady-state friction coefficient is different for different materials that were subjected to an external vibration [68]. Chowdhury et al. [68,69] carried out detailed studies

on the influence of vibration amplitude and frequency on the friction of different polymer/polymer composites. Figure 17 shows the influence of the external vibration on the friction behavior of PTFE and rubber. It is clear that the friction coefficient declined with the rise of vibration amplitude. The higher the vibration amplitude, the longer is the actual friction time due to more detachment between mating surfaces under vibration [68–70]. With an unchanged frequency, the increase of vibration amplitude tends to decrease the transient perpendicular load, which reduces the actual normal load and thus friction coefficient [70]. This has been experimentally verified by Chowdhury et al. [68,69]. The factors accounting for this temporarily load are: (a) the superposition of static and dynamic forces produced from vibration, (b) the change of friction vector, (c) the localized transition of vibration energy into heat energy, and (d) the limiting excitation frequency to resonance frequency.

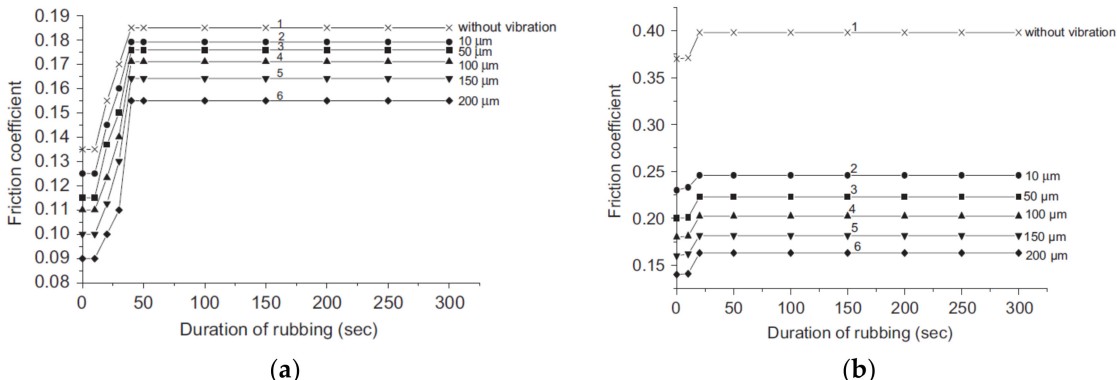

**Figure 17.** Change of friction coefficient as a function of rubbing time and amplitude of vibration: (**a**) polytetrafluoroethylene (PTFE) and (**b**) rubber [68].

The influences of vibration amplitude on friction coefficient are shown in Figure 18. As pointed out by Chowdhury et al. [68,69], the reduction in the coefficient of friction is almost linear for rubbers and polymer composites. However, this is not true for PTFE, which increased with the rise of vibration amplitude [68–70]. The main reason behind that is the inherent viscoelastic behavior of rubbers, which is capable of absorbing more energy when compared to PTFE. The variation of the friction coefficient with vibration also relies on various physical properties of sliding materials and adhesive properties, which are related to surface free energy, inter-atomic force, Vander Waals forces, interface state, and chemical reactivity. Experiments on various materials at unlike frequencies of vibration showed that the time to achieve constant friction was identical for those materials [68–70].

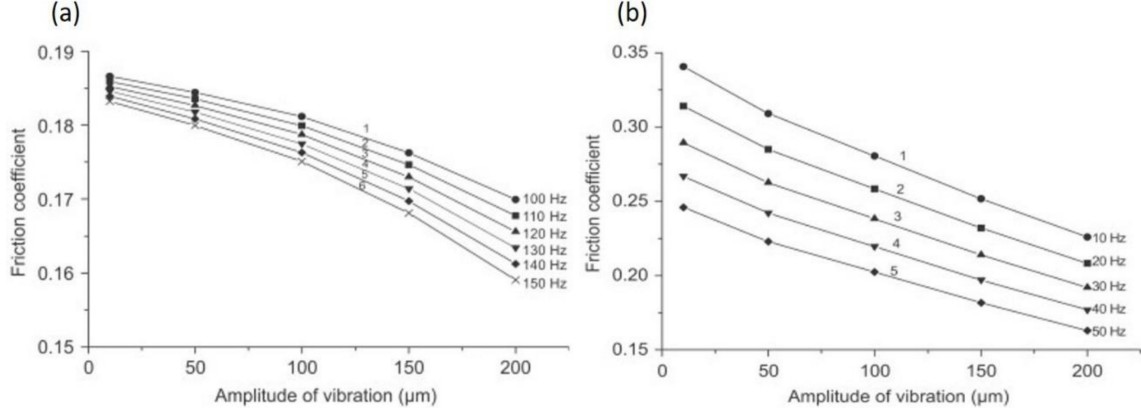

**Figure 18.** Change of the coefficient of friction on the different material at different vibration amplitude with various frequencies: (**a**) PTFE and (**b**) rubber [70].

However, in the case of horizontal vibration, the scenario was opposite in the case of glass fiber incorporated polymer (GFRP), as reported by Chowdhury et al. [68,69] (cf. Figure 19). In this

case, friction coefficient increased approximately linearly when the amplitude of horizontal vibration increased. The rise of the coefficient of friction [70] is related to: (a) the variation of inertia force along friction force, (b) a greater abrasion shearing because of dissemination as well as ploughing of asperities among mating surfaces, (c) the micro welding of contact asperities, which reverses to friction vector and causes mechanical joining, and (d) the increase in material deformation due to high temperature. In addition to the above-mentioned factors, it is also possible that there is a replenishment of transfer films/layers during the process. Thus, the wear and friction of polymer/polymer composites with the vibrations in the longitudinal direction is greater than those in the transverse direction, due to the inertia forces of the vibrating bodies. Henceforth, the friction force between mating components would rise due to the rise of natural frequency [68]. The increase in friction force, in turn, accelerates the wear rate for a number of reasons, such as: (a) the higher ploughing and (b) more damage of surface and rupture of incorporated particles [71]. Unfortunately, there is no report available in literature regarding the influence of amplitude and frequency on wear and friction behavior of HPPs so far, which will be an exciting field to explore. It is also important to note that vibration does commonly exist in sliding systems and does not necessarily have to be imposed by external sources.

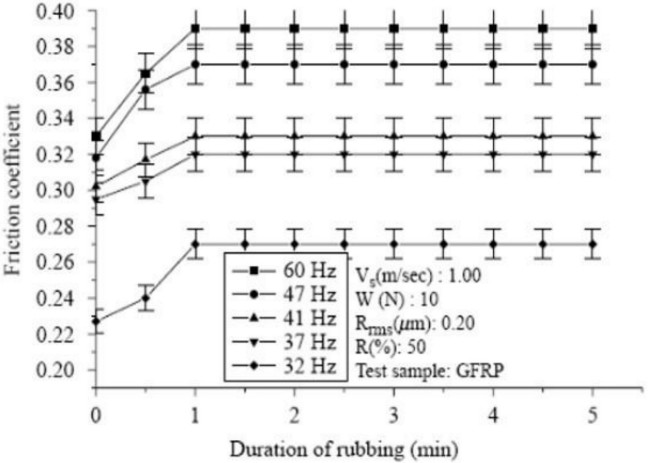

**Figure 19.** Change of friction coefficient due to changes at the longitudinal direction of vibration on polymer composite (GFRP) [70].

## 4. Correlation between Mechanical and Tribological Properties

Although there is no a simple relationship between the basic mechanical properties and tribological performance, it is important for materials engineers and scientists to understand the dependence of the friction and wear behavior on intrinsic material parameters, such as the type of fillers and basic mechanical properties. Such an understanding is critical for industries to select and design new, high-performance polymeric materials. According to commercial manufacturers (Ensinger, Germany; Solvay Advanced Polymers, USA), HPPs, such as PEEK and PPP, possess quite extraordinary mechanical properties, as compared to other commonly used polymers (Figure 20). Both modulus and strength are much higher than that of thermosetting polyimide (PI). Particularly, in case of compressive characteristics, the strength of PPP under compression surpasses other polymers (PEEK, PC, PPSU) by a factor of 5. At room temperature, PPP provides a significant possibility when other high performing thermosets and thermoplastics stretch to their mechanical properties' boundaries. However, by considering the effect of the service environments, such as temperature, this is not the case in all ways. Recently, Kurdi et al. [98] reported that, though the tribological behaviors of PPP at room temperature are promising, its high-temperature aspect is questionable, as shown in Figure 21. The wear resistance of all the tested polymeric samples decreased with the increase of environment temperature. Nevertheless, it was the worst in the case of PPP. This may be due to the integral nature of PPP, as indicated by K. Friedrich et al. [28], in view of their amorphous nature

through TGA-DSC analysis. PPP shows a broad temperature assisted swelling rather than a sharp melting point which is mostly true for all thermoplastics [101–104]. Therefore, when the material is subjected to friction, generated heat at contact soften the amorphous polymer. The wear debris would come off in layer by layer (instead of fragmented particles) with a high wear rate. Thus, PPP, a mechanically sound and superior polymer, fails to retain its tribo-response at high temperature. In general, the better the mechanical characteristics (such as hardness) of the materials, the lower is the rate of wear, as materials with the higher mechanical property will show more resistance against wear debris formation. However, there is no one-to-one relationship between the hardness of the material and its wear rate, as wear is a system response rather than material property. This observation also holds in the present case, as PPP is the hardest material among the common polymers (Figure 21).

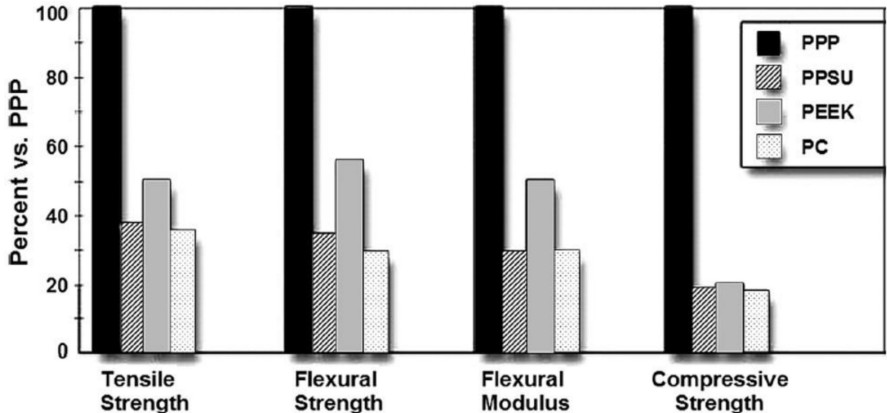

**Figure 20.** Ranking of different HPPs based on mechanical properties [3].

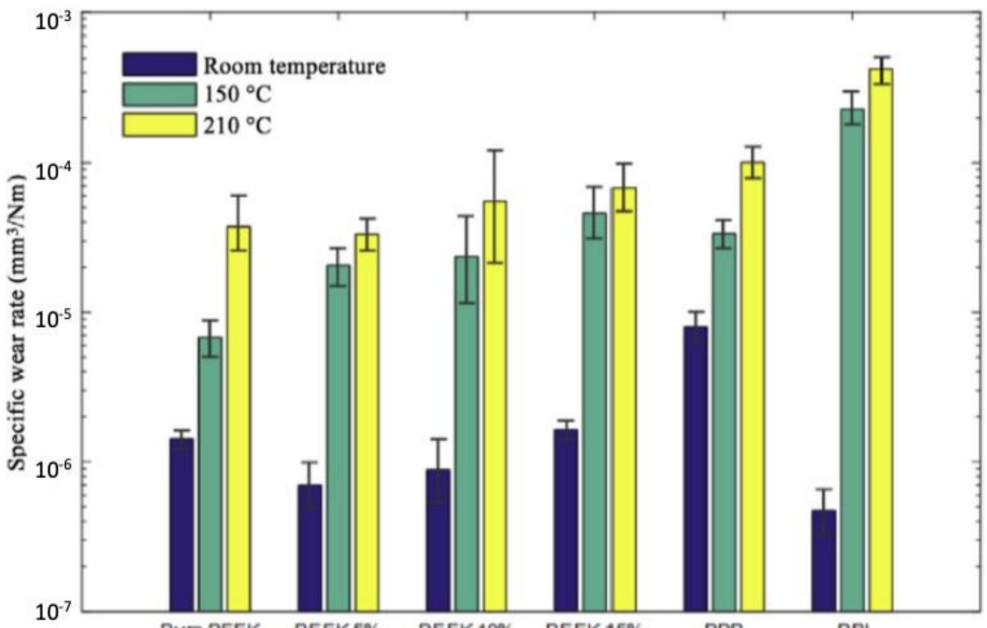

**Figure 21.** Wear rate of high-performance polymers under different temperature regime sliding against steel disk under pin-on-disk test configuration [98].

Nevertheless, continuing efforts have been made to understand the dependence of wear on other mechanical properties, e.g. for the design of wear resistant polymer composites using various fillers. In general, though the introduction of fillers in a polymer causes the discontinuities of material to some extent, its melting temperature and extent of crystallinity are mostly unchanged, as reported by Friedrich et al. [34]. Therefore, the thermal behavior of the polymer matrix in the composite appears to vary slightly from the equivalent neat polymer. The incorporated fillers usually increase

the mechanical characteristics of the composites, for example, tensile strength, hardness, elongation at break, and flexural modulus. With the enhanced mechanical performance, it is expected that the composite could experience less wear rates, which, however, requires a good understanding of the wear mechanism, associated with the failure mechanics in materials under the given sliding condition. For example, the expression for lubricated circumstances was theoretically formulated from the perceptions of crack progression, damage build-up, and traditional fracture mechanics, as reported by Lhymn et al. [89]. Accordingly, the rate of wear is inversely proportionate to the product of the elastic modulus (E), hardness (H), and elongation at break (ε), as shown in Equation (7) [105,106]:

$$wear\ rate\ \propto\ \frac{1}{H.E.\varepsilon} \tag{7}$$

Basak et al [16] has applied the equation to describe the wear behavior of PEEK composites filled various fillers, as shown in Figure 22. As the content of nanoparticle ranges 5–20 wt. %, a linear relation to *1/H.E.ε* was observed. The wear resistance of the hybrid composites (where both fibre and particle incorporated) tends to increase with the rise of $ZrO_2$ content in the range from 5 to 15 wt. % Subsequently, it displayed a solid drop-off trend when the content of $ZrO_2$ was 20 wt. %, which is exceeding the critical incorporate content. A combination of differing types of filler showed different influences on mechanical properties as fiber type fillers could significantly enhance hardness and modulus. However, the tensile strain at break reduced significantly. Therefore, *H.E.ε* might be controlled by varying the amounts of different fillers of hybrid composites to achieve the highest possible values of *H.E.ε* [16]. Additional research must be dedicated to find further details on how the mechanical characteristics influence the wear behaviors when a combination of different types of fillers are in use. Further, the external sliding condition may also affect such a relation between wear and mechanical properties.

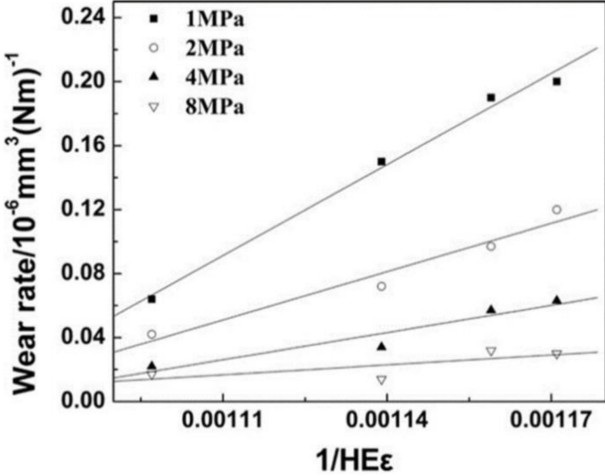

**Figure 22.** Influence of mechanical properties on the wear rate of the hybrid PEEK [16].

Recently, Hakami et al. [107] investigated the influence of the various size of abrasive particles on the wear characteristics of different polymer samples. By comparing the effect of mechanical properties on wear rate subjected to abrasive particles with different sizes, it was observed that, with a small particle size (125 μm), the wear is more influenced elongation at break and tensile strength more. In contrast, bigger abrasive particles (425 μm) could cause larger penetration and result in a higher wear rate. Moreover, the generated wear particles of polymers with lower hardness tend to clog abrasive's surface and increase the real contact area, leading to a further increase in wear rate [39,108]. In view of that, polymers' hardness and tear strength are dominant mechanical properties under such conditions.

## 5. Conclusions and Future Directions

The present work discusses the aspects of tribological properties of HPPs and their composites. The wear performance of materials highly depends on the engineering system, where it should be used. With the intention of understanding the sliding wear mechanisms of HPPs and their composites, two factors should be taken into account: (i) materials and (ii) system parameters, such as environment temperature, vibration, and lubrication conditions.

In practice, HPPs are commonly used by incorporating with different types of micro/nanofillers, as well as their combinations. The fillers such as inorganic particles can improve the wear resistance either due to enhanced mechanical characteristics (such as hardness, modulus, and stiffness) or by forming durable and resilience transfer film layer between contact materials. However, the addition of fillers may also cause discontinuities in the polymer matrix and be susceptible to generate wear debris. The latter can also act as the third body, which sometimes tends to increase wear of the system. In that respect, there is a critical filler content beyond which the addition of filler actually degrades the wear resistance of the materials. The critical filler content varies from material to material, however roughly in the range of 5–15 vol. %, as reported in literature. A topographic smoothening and a probable rolling action of the nanoparticles are possible explanations for their improvement on the wear and friction performance. There is also a trend of using a mixture of particle and fibre type fillers together to improve the performance of the polymer composites. In that case, wear mechanism can be more complex.

In summary, the applications of polymer composites in tribological fields are broad and growing in the quest of increasing efficiency and lowering operating cost. In the last few years, particular attention has been paid to HPPs and their composites for engineering fields under extreme conditions, where lower friction and lower wear ought to be achieved. Additional encouraging impacts on the accomplishment of structural tribo-components should be anticipated from the advancement of functionally gradient materials and the hybrid polymer composites with various fillers. For designing future HPP composites, the following factors should be considered: (i) the compatibility of the fillers with the matrix, (ii) a blend of fibers and particulate fillers, and (iii) a positive synergic of the fillers with counter surfaces. The main intention will be forming and maintaining effective TLs during the sliding. There is still sufficient space for the future growths of HPPs and their composites as pointed out hereafter:

1. The recent trend towards the development of HPPs is to use the combination of particle and fibre type reinforcements. This technique offers the possibility to increase both strength and toughness of HPPs, though makes the scenario more complex in terms of wear mechanism and wear-debris formation. In that respect, a more fundamental investigation is needed in terms of their interactions and the overall effect on friction and wear behavior.

2. Effects of vibration, both natural and system generated, is another under-research area in this field. Vibration may change the contact mode in the mating materials and accelerate/decelerate friction and wear accordingly. Proper care should be taken in that respect, which has been often overlooked.

3. There are some contradictions in understandings regarding high-temperature tribological aspects of HPPs and their composite, as some authors reported that the incorporation of particles does not necessarily increase the wear performance of HPPs composite, particularly at high temperature. In addition to that, consideration will be taken regarding the cost of nanofiller and their relatively weighted advantages.

**Author Contributions:** A.K. wrote the manuscript under the supervision of L.C. L.C. has developed the idea and edited the article.

**Funding:** This research received no external funding.

**Acknowledgments:** Abdulaziz Kurdi gratefully acknowledges to King Abdulaziz city for Science and Technology (KACT) for financial support for this research through scholarship.

## Abbreviations

**Acronyms**

| | |
|---|---|
| HPPs | High performance polymers |
| TFL | Transfer film layer |
| TL | Transfer layer |
| PET | Polyethylene terephthalate |
| CST | Continuous service temperature |
| PSU | Poly sulfone |
| PES | Poly ether sulfone |
| PVDF | Poly vinyl dene fluoride |
| PEI | Poly ether imide |
| PPS | Poly phenylene sulphide |
| PEEK | Poly ether ether ketone |
| SBC | Styrene butadiene copolymers |
| PK | Poly ketone |
| PEK | Poly ether ketones |
| PPP | Poly para phenylene |
| PBI | Poly benzimidazole |
| PTFE | Poly tetra fluoro ethylene |
| GFRP | Glass fibre reinforced polymer |
| PI | Poly imide |
| PTW | Potassium Titanate Whisker |
| EP | Epoxy polymer |
| PPA | Polyphthalamide |
| TPI | Thermo plastic polyimide |
| PPSU | Poly phenyl sulfone |
| PC | Poly carbonates |
| DSC | Differential scanning calorimetry |
| FIB | Focused ion beam |
| SEM | Scanning electron microscopy |
| TEM | Transmission electron microscopy |
| XRD | X-ray diffraction |
| TGA | Thermogravimetric analysis |
| DSC | Differential scanning calorimetry |
| DMA | Dynamic mechanical analysis |
| Gr | Graphite |
| SCF | Short carbon fibres |

**Symbols**

| | |
|---|---|
| $p$ | Pressure |
| $v$ | Sliding velocity |
| $T_g$ | Glass transition temperature |
| $Si_3N_4$ | Silicon nitride |
| $SiO_2$ | Silicon dioxide |
| $\alpha$-FeOOH | Goethite |
| $ZrO_2$ | Zirconium dioxide |
| $TiO_2$ | Titanium dioxide |
| $W_t$ | Time-related depth of wear rate |
| $k^*$ | Wear factor |
| $t$ | Test duration |
| $\Delta h$ | Height loss of the specimen |
| $\varphi$ | Bearing modulus |
| $\delta S$ | Solubility parameters of fluid |
| $\delta P$ | Solubility parameters of polymer |
| $T_m$ | Melting temperature |

| | |
|---|---|
| $H_c$ | Composite hardness |
| $A_t$ | Total projected indentation contact area |
| $A_f$ | Portions of projected contact areas in film |
| $A_s$ | Portions of projected contact areas in substrate |
| $H_f$ | Intrinsic harnesses of film |
| $H_s$ | Intrinsic harnesses of substrate |
| $h_t$ | Total indentation depth |
| $h_f$ | TL thickness |
| $h_s$ | Indentation depth in substrate |
| $\lambda$ | Transfer film efficiency factor |
| $R_a$ | Surface roughness |
| $H$ | Hardness |
| $E$ | Young's modulus |
| $\varepsilon$ | Elongation at break |

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
