# Peer review of "Recent Advances in High Performance Polymers—Tribological Aspects"

_lubricants, doi:10.3390/lubricants7010002_

Round 1

Reviewer 1 Report

Summary and Recommendation:

The article raises an interesting topic about the tribological properties of high performance polymers. This extensive study is well defined and focuses on the topic. It can be accepted for publication after minor revision.

The article requires the English language edition.

Friction coefficient and the thermal properties of all studied polymers (Tg, Tm, HDT) should be summarized in a form of a table.

The article contains many inaccuracies related to polymers.

Specific comments:

The abbreviation HHP lines 28,31 ect. is incorrect.  Previously, HPP was used.

Line 37 – “high performance epoxy” requires specification.      

Line 133 – I suggest using another adjective for  "normal polymer"

Line 124 – what do you mean by saying that PET is a “multifunctional polymer”? Do you refer to applications,  mechanical performance or chemical reactivity? In the polymer chemistry, the  term functional polymer refers to chemical reactivity. Thus, it should be replaced by another one.

Line 131, 158, 159 – terms: thermal stability,  heat stability has to be specified. Normally, thermal stability is characterized by HDT. What do you mean?

Line 164 – “Polyparaphenylene” is wrongly    written. Poly(p-phenylene) is      correct. The names of all polymers mentioned in the work should be checked      and, if necessary, corrected.

Line 177 – it is not “compounding”, which refers to  physical process. In the case of cross-linking, this term refers to   compounding substrates before hardening. Later, they are chemically  incorporated into the network. This sentence has to be revised.

Lines 179 – 181 – the sentence requires citation.  I recommend to revise this sentence as well.

Line 742-743 – can you support this sentence with      a figure? Typically, the decrease in storage modulus as a function of      temperature is small. However, a rapid decrease in E’ takes place, when a      polymer transforms from glassy state to viscoelastic state. Which decrease      do you mean?

Line 755 – the sentence is confusing.

Figure 14 – “e” is lacking in the word “specific”.

Reviewer 2 Report

The article is a review of the research presented in the literature. The number of analyzed works is large (over 100 literature items). They are related to the subject of the article and reasonably up to date. The reviewed paper has rather the "state-of-art of tribology of high-performance polymers" nature than the paper about tribological processes during lubrication.

The Authors describe some examples of tribological properties of polymers during friction in the presence of lubricants, but only a few described problems strictly related to lubrication. In the case of publication of the article in the"Lubricants" journal, more attention should be paid to lubrication processes during the friction of polymeric materials. For example, you can skip fragments about dry friction (e.g. temperature effect). Instead, the text of the article could be supplemented with a description of the phenomenon of adhesion between lubricants and polymers and surface wettability of polymeric materials by lubricants (water, oils etc.). This is important for the description of the lubrication process of machine parts made of polymers.

The content of the article in its present form is poorly connected with the aims and scope of the "Lubricants" journal. (See: Website of the journal).

The text could be published in the "Lubricants" journal after its correction and additions or could have been sent to the "Polymers" journal if it possible. I think that the better solution is publishing that paper in the "Polymers" journal (MDPI Open Acces).

Reviewer 3 Report

In their work, Kurdi et al have reviewed the tribological properties of high performance polymers. The review is quite topical and well written. I would like the authors to consider the following points:

Throughout the manuscript, the abbreviation for HPP is sometimes being written as HHPs. Please correct that.  

Line 48, regarding the PTFE composite in reinforcement, I would like the authors to consider the following works which provide evidence for surface functionalisation of PTFE powders for reinforcement in HPPs. 

Hunke et al, Wear, 328, pp 480-487 (2015); Hunke et al, Wear, 338, pp 122-132 (2015)

Line 140:        ".....polymers dont accumulate electrical charges". The statement is incorrect as the polymers are used as dielectric materials and have been shown in triboelectric nanogenerators. 

Line 232, 235: Please re-write the sentences to make them grammatically correct. 

Line 403, reconsider the language of the sentence and its readability 

Fig.6, the quality of the image can be improved

Line 661, spellcheck "polifenialamid"

Line 680: Please provide the full form for "PPP"

In the conclusions and future work section, there seems to be some repetition especially for point 1 which seems to have been discussed already in the preceding paragraph. Please correct that.  

Round 2

Reviewer 1 Report

The article can be accepted for publication. However, I suggest two minor corrections.

Lines 159 -162 – the polymer nomenclature should be re-reviewed, for example: Poly(vinylidene fluoride), Poly(p-phenylene) (p – should be written in italics).

Lines 171 – 174 – the sentence is still unclear. By using multifunctional comonomers (substrates with functionality greater than 2) during copolymerization, additional functionalities may be introduced into the macromolecule. They can undergo further reaction with the crosslinker. 

Author Response

Please find the replies to the reviewer comments as attached.

Reviewer 2 Report

After receiving explanations from the Authors of the article and the Assistant Editor, Ms. Jasmine Xu, I accept the article in the new form presented. I believe that in its current form the article will be in compliance with the focus of the special issue "Advances in Polymer Tribology".

Author Response

Please find the replies to the reviewer comments as attached
